# Pathogenesis of Warthin’s Tumor: Neoplastic or Non-Neoplastic?

**DOI:** 10.3390/cancers16050912

**Published:** 2024-02-23

**Authors:** Ryogo Aoki, Takuji Tanaka

**Affiliations:** Department of Diagnostic Pathology (DDP) & Research Center of Diagnostic Pathology (RC-DiP), Gifu Municipal Hospital, 7-l Kashima-Cho, Gifu City 500-8513, Gifu, Japan; raoki@gmhosp.gifu.gifu.jp

**Keywords:** Warthin’s tumor, pathogenesis, neoplastic, SASP, senescent cells, DNA damage, inflammation

## Abstract

**Simple Summary:**

Although Warthin’s tumor is a well-known and frequent tumor in the salivary gland, its pathogenesis is not fully understood. Warthin’s tumor is composed of oncocytic epithelial cells lining papillary and cystic structures in a lymphoid stroma. Previously several hypotheses have been postulated. The risk factors for its development are known and they include aging, smoking, and radiation exposure. Recent findings have suggested that chronic inflammation and aging cells promote the growth of Warthin’s tumor. In this short review, we propose that DNA dame, metabolic dysfunction of mitochondria, senescence-associated secretory phenotype, human papillomavirus, and IgG-4 may be involved in the development of Warthin’s tumor.

**Abstract:**

Warthin’s tumor is the second most frequent neoplasm next to pleomorphic adenoma in the salivary gland, mostly in the parotid gland. The epithelial cells constituting a tumor are characterized by the presence of mitochondria that undergo structural and functional changes, resulting in the development of oncocytes. In addition to containing epithelial cells, Warthin’s tumors contain abundant lymphocytes with lymph follicles (germinal centers) that are surrounded by epithelial cells. The pathogenesis of Warthin’s tumor is not fully understood, and several hypotheses have been proposed. The risk factors for the development of Warthin’s tumor, which predominantly occurs in males, include aging, smoking, and radiation exposure. Recently, it has been reported that chronic inflammation and aging cells promote the growth of Warthin’s tumor. Several reports regarding the origin of the tumor have suggested that (1) Warthin’s tumor is an IgG4-related disease, (2) epithelial cells that compose Warthin’s tumor accumulate mitochondria, and (3) Warthin’s tumor is a metaplastic lesion in the lymph nodes. It is possible that the pathogenesis of Warthin’s tumor includes mitochondrial metabolic abnormalities, accumulation of aged cells, chronic inflammation, and senescence-associated secretory phenotype (SASP). In this short review, we propose that DNA damage, metabolic dysfunction of mitochondria, senescent cells, SASP, human papillomavirus, and IgG4 may be involved in the development of Warthin’s tumor.

## 1. Introduction

Warthin’s tumor, also known as papillary cystadenoma lymphomatosum, monomorphic adenoma, or adenolymphoma, is the second most common tumor of the parotid gland after pleomorphic adenoma, accounting for approximately 15% of all parotid tumors, and is encountered relatively frequently in daily clinical practice [1,2]. An investigation by Franzen et al. [3] suggested that Warthin’s tumor was the most common histological type in the period from 1997 to 2017. The researchers also suggested a growing incidence in women and a decreasing age of patients [3]. The site of occurrence is restricted to the parotid gland and surrounding lymph nodes, with a high frequency of simultaneous or ectopic multiple or bilateral occurrence [4,5]. The tumor usually presents as a painless mass, but it may be painful when the lesion is associated with inflammation [6]. Clinically, an ultrasound examination reveals the tumor as an oval and well-defined mass with multiple anechoic areas, or an anechoic mass with posterior acoustic enhancement. Rapid growth is stimulated by infection. In some cases, multiple septa and intra-tumoral fluid thickness can cause non-uniform echo patterns [7,8,9]. Treatment is usually based on surgical resection; however, most patients have low malignancy rates. Therefore, conservative treatment may be an option if Warthin’s tumor is preoperatively diagnosed [10,11].

Although it has been more than 100 years since the discovery of Warthin’s tumor, the etiology remains unclear [12]. In this short review, we propose a possible association between DNA damage, metabolic dysfunction of mitochondria, senescent cells, senescence-associated secretory phenotype (SASP), human papillomavirus, and IgG4, which may be involved in the development of Warthin’s tumor. Our hypotheses as described support the results of the investigation by Kuzenko et al. [2].

## 2. Anatomy/Histology of the Salivary Glands

The salivary glands are exocrine glands that produce saliva. Humans have three major paired salivary glands (parotid, submandibular, and sublingual glands) as well as numerous minute salivary glands. The salivary glands can also be classified according to their secretions as serous, mucous, or seromucous (mixed). In serous secretions, the main type of secreted protein is α-amylase, an enzyme that breaks down starch into maltose and glucose, whereas in the mucous glands, mucin, which acts as a lubricant, is the main secreted protein. In humans, saliva is produced every day. The secretion of saliva (salivation) is mediated by parasympathetic stimulation. Acetylcholine is an active neurotransmitter that binds to muscarinic receptors in the glands, leading to increased salivation.

The working parts of salivary glandular tissue consist of secretory end pieces (acini) and a branched ductal system (Figure 1). In serous glands (e.g., parotid glands), the cells in the end piece are arranged in a roughly spherical shape. Mucous glands tend to be arranged in a tubular configuration with a larger central lumen. In both types of glands, the cells in the end piece surround the lumen, which is the start of the ductal system (Figure 1). Three types of ducts are present in most of the salivary glands. The fluid first passes through the intercalated ducts, which have a low cuboidal epithelium and a narrow lumen. From there, the secretions enter the striated ducts, which are lined with columnar cells with many mitochondria. Finally, the saliva passes through the excretory ducts, where the cell type is cuboidal, until the terminal part, which is lined with a stratified squamous epithelium. The end pieces may contain mucous cells, serous cells, or a mixture of both. A salivary gland can consist of a varied mixture of these types of end-pieces. In mixed glands, mucous acini are capped by a serous demilune. In addition, myoepithelial cells surround the end piece, and their function is to assist in propelling secretions into the ductal system. The gland and its specialized nerves and blood supply are supported by connective tissue stroma.

## 3. Histopathology of Warthin’s Tumor

The histopathology of Warthin’s tumor is defined by the tubular, cystic, and papillary proliferation of highly cylindrical oncocyte-like cells with eosinophilic granular sporulation. It is well demarcated from the surrounding normal salivary gland tissue [13,14]. The tumor is also characterized by a biphasic arrangement of similar oncocyte-like cuboidal cells with cylindrical cells on the basal side. The stroma is occupied by mature, non-atypical small lymphocytes with lymph follicles (germinal centers), although the number of stroma varies from case to case [2] (Figure 2a). This may be caused by an immune response to the tumor epithelium or by residual lymphoid tissue within the lymph nodes that is partially replaced by the tumor epithelium [15]. In addition, the cytoplasm of cells exhibiting oncocytes possesses an excessive accumulation of mitochondria [16]. This may be a result of the accumulation of senescent mitochondria due to the reduction in cellular mitophages and is associated with a deletion of 4977 bp in the mitochondrial genome [17,18,19,20]. It is not uncommon for intermingled goblet cell-type mucous cells, glandular hairy metaplastic cells, and squamous metaplastic cells to be observed. Basal cells do not usually differentiate into myoepithelial cells. The cystic structure, lined with epithelial cells, is filled with necrotic contents, including cholesterol crystals. When the cystic structures rupture, the fluid leaks into the lymphangitic stroma, causing epithelioid granulomas with neutrophil infiltration. In association with this, squamous epithelium and mucous cells sometimes appear due to inflammatory or chemogenic changes, and in rare cases, extensive necrosis may accompany the lesions. Neither the epithelial nor the lymphocytic component is usually atypical; however, secondary adenocarcinoma, mucoepidermoid carcinoma, squamous cell carcinoma, oncocytic carcinoma, or malignant lymphoma (follicular lymphoma) may occur [21,22,23,24,25,26,27,28]. The most common histologic type of carcinoma derived from Warthin’s tumor is squamous cell carcinoma, although mucoepidermoid carcinoma has also been reported [29,30].

## 4. Cytology of Warthin’s Tumor

A cytological diagnosis of Warthin’s tumor can be made by observing a two-cell pattern of lymphocytes and epithelial cells, with lymphocytes in the background and eosinophilic cells in clusters or sporadically isolated. The aggregates of oncocytes range in size from large to small sheets without abnormal overlapping [31]. On Papanicolaou staining smears, oncocytes appear with light green stained granular cytoplasm, often with eosinophilic changes that stain orange G. Their nuclei are small and slightly atypical, sometimes with a few small nucleoli, but they are often obscure. Occasional findings include cystic structures, including oncocytes floating in the lumen (Figure 2b). When cyst contents of punctured cells contain hypercylindrical nucleated oncocyte cells along with histiocytes, the diagnosis of Warthin’s tumor is inferred; however, in the absence of nucleated oncocyte cells, the diagnosis is more challenging [32]. In May–Giemsa-stained specimens, Warthin’s tumor cells do not possess a metachromatic component, and in some cases, there are various mature lymphocytes in the background.

The cellular presentation of squamous metaplastic Warthin’s tumor, which is a secondary alteration of Warthin’s tumor, is a mixed appearance of neutrophils and histiocytes with orange G-stained metaplastic squamous cells [33]. In addition, mucous cells often appear with metaplasticity, in which case mucous cells are interspersed with collections of oncocytes on a mucous background. For the diagnosis of Warthin’s tumor, fine needle aspiration cytology (FNAC) is useful in the preoperative diagnosis of salivary gland tumors because it is minimally invasive and has many typical cytological findings [34,35,36]. Data have been reported that FNAC has a sensitivity of 93%, a specificity of 94.8%, and an accuracy of 94.6% in the diagnosis of Warthin’s tumor [36].

## 5. Risk Factors for Warthin’s Tumor

Warthin’s tumors were first reported over 100 years ago, but their pathogenesis is not fully understood [12]. Various possible pathogeneses and risk factors have been described. (1) The majority of Warthin’s tumors show an obvious marginal sinus beneath the tumor capsule and have an intralymphatic origin or a metaplasia of normal salivary gland epithelial or ductal cell origin [37,38,39,40]. (2) Catalytically inactive glyceraldehyde-3-phosphate dehydrogenase (GAPDH) was found to bind to damaged mitochondria and incorporate these mitochondria directly into lysosomes, exhibiting a characteristic immunohistochemical GAPDH staining pattern in Warthin’s tumor cells, suggesting either whole cell progressive loss of cytoplasmic GAPDH (Figure 3), likely due to loss or nuclear shift of the protein [18]. (3) The epithelium of Warthin’s tumor, whether hyperplastic, metaplastic, or neoplastic, interacts with lymphoid tissue [2]. (4) A high association with smoking, which causes chronic inflammation of the epithelium, has been reported [41,42,43]. (5) The incidence of Warthin’s tumor is reported to be high after radiation exposure [44]. (6) Patients with Warthin’s tumor have been reported to show an increased incidence of autoimmune or infectious diseases [12,45]. (7) It has been reported that angiogenesis and lymphangiogenesis are increased, reactive lymphocyte hyperplasia is induced, and that the two elements, epithelial cells and lymphocytes, are not simply present by chance but are interdependently related to tumor development [46]. (8) HPV infection is reportedly associated with the development of Warthin’s tumor [47]. (9) IgG4-reated disease (IgG4-RD) has been reported to be indirectly involved in the development of Warthin’s tumor [48]. To date, there is no consensus on the development of Warthin’s tumor. However, based on our previous studies, we believe that mitochondrial metabolic abnormalities, senescent cell accumulation, chronic inflammation, SASP, and HPV infection may be involved in the pathogenesis of Warthin’s tumor.

## 6. Association between SASP and Warthin’s Tumor

Normal somatic cells irreversibly arrest the cell cycle after a certain number of divisions. This is caused by telomere shortening, also known as replicative senescence [49]. The same phenomenon occurs in normal epithelial cells with proliferative capacity when they receive carcinogenic stimuli such as DNA damage, oxidative stress, or excessive proliferative stimulation by oncogene (Ras) products. This is called “premature senescence”, which does not involve telomere shortening. This phenomenon is considered to be a mechanism of cancer suppression. It has been reported that the proportion of senescent cells increases in tissues and organs with aging, and cellular senescence is also involved in the pathogenesis of individual aging and age-associated diseases. The process of cellular senescence is as follows: telomere shortening, radiation, carcinogens, and oxidative stress trigger the DNA-damage response (DDR), which activates the p53 pathway and induces cellular senescence. Double-stranded DNA damage arrests the cell cycle and proceeds to the repair process, whereas irreparable DNA damage induces apoptosis and cellular senescence. DDR involves kinases such as ataxia telangiectasia mutated (ATM) and checkpoint-2 (CHK2), adapter proteins (e.g., 53BP1, mediator of DNA damage checkpoint protein-1 [MDC1]) and chromatin-modifying proteins (e.g., phosphorylation of histone H2AX [γ-H2AX]) (Figure 4), many of which are localized to sites of DNA damage [50]. When the p53 pathway is inhibited and the cell cycle continues to progress in senescent cells, the telomere length continues to decrease. Eventually, the loss of telomeric DNA causes severe genomic instability, leading to cell death, known as mitotic collapse. Cell cycle arrest activates the p53-p21 and p16-retinoblastoma (RB) pathways [51,52]. While senescent cells usually remain in the G1 phase of the cell cycle, their intracellular metabolism is active and characterized by protein secretion phenomena, including, for example, inflammatory cytokines (e.g., interleukin [IL1]β (Figure 5), IL6 (Figure 6), IL8, plasminogen activator inhibitor [PAI]-1, vascular endothelial growth factor receptor [VEGF], matrix metalloproteinase [MMP]3, etc.), which are called “senescence-associated secretory phenotypes” (SASPs) [53,54]. SASPs have been shown to promote the development of age-related diseases, induce oncogenesis/carcinogenesis, and increase tumor size while enhancing tissue repair and regeneration [55,56,57,58]. For example, senescent fibroblasts and many other SASP factors enhance cancer cells’ proliferation and invasion in culture systems [59,60]. Thus, cellular senescence has a dual nature: inhibiting tumors while promoting them [57,61,62,63].

Chronic inflammation is closely associated with tumorigenesis [64,65]. In addition, age-associated cellular senescence is thought to act as a tumor promoter by initiating several inflammatory processes. Chronically activated leukocytes produce direct and indirect mitogenic growth factors (epidermal growth factor [EGF], tumor growth factor [TGF]β, tumor necrosis factor [TNA]α, fibroblast growth factor [FGA], interleukin [ILs], chemokines, histamine, and heparin), which stimulate tumor and stromal cell proliferation. In addition, inflammatory cells such as macrophages, granulocytes, monocytes, and mast cells secrete diverse classes of proteolytic enzymes that modify the structure and function of the extracellular matrix (ECM) and release mitogenic factors. Macrophages also produce vascular endothelial growth factor (VEGF) and EGF when exposed to T helper 2 (Th2)-type cytokines, such as IL4, which promote angiogenesis and metastasis. In addition, Th2 cells are well recognized as tumor promoters. Th2 cells are “driven” by OX40 ligand (L)-expressing dendritic cells in response to cancer-derived thymic stromal lymphopoietin (TSLP) [66]. Th2 CD4^+^ T lymphocytes secrete IL4 and IL13. Subsequently, macrophages release EGF, VEGF, and TGFβ to promote tumorigenesis [67,68].

## 7. Mitochondrial Dysfunction and Warthin’s Tumor

Warthin’s tumor is morphologically composed of oncocytic epithelial cells with abundant papillary and cystic mitochondrial structures in the lymphoid stroma. Mitochondria are intracellular organelles that synthesize ATP using high-energy electrons and oxygen molecules. In addition, mitochondria actively fuse and divide to stabilize their morphology. Recently, interesting findings showing that (1) “mitochondrial DNA mutations accumulate in human tissues with senescence” [69]; and (2) “mitochondrial dysfunction induces aging” have been reported [70]. With aging, mitochondria become highly susceptible to morphological changes, and these changes lead to reduced function due to oxygen radical damage, ultimately causing the organism to age [71]. Since mitochondria are the primary source of cellular ATP and are involved in the biosynthesis of deoxyribose nucleoside triphosphate (dNTP), mitochondrial dysfunction results in a reduction in ATP levels and alterations in ATP-dependent pathways that are involved in transcription, DNA replication, DNA repair, and DNA recombination. Additionally, mitochondrial defects may lead to mutagenesis of the nuclear genome [72]. Approximately 10% of mtDNA has a “common” 4977 bp deletion. One study, in which polymerase chain reaction (PCR) was used to further quantify 4977 bp deletion in normal parotid control tissue that was age-matched to Warthin’s tumor, revealed that deletions were present in all parotid tissues, but the changes were significantly greater in oncocytic tumors. Although there were a small number of controls, there was a tendency towards higher concentrations of deletions in smokers [73].

Recently, a group of hydrogen peroxides that function in mitochondrial fusion and fission was identified. Mitofusin (Mfn) 1, Mfn2, and OPA1 are involved in fusion, while dynamin-related protein 1 (Drp1) (as well as dynamin-like protein [DLP1]) is involved in fission [74]. Mitochondrial mitosis is associated with mechanisms that promote cell cycle arrest and apoptosis. In the cell cycle, Drp1 Ser 616 is mainly phosphorylated in the early S phase, which leads to the promotion of mitochondrial division and movement of cells into G2/M. Loss of Drp1 induces mitochondrial hyperfusion, leading to ATM-dependent G2/M arrest and apoptosis. The overexpression of Drp1 has been associated with malignant oncocytic thyroid tumors, and genetic and pharmacological blockade of Drp1 activity has been reported to affect the migration and invasion of thyroid cancer cells, which is a characteristic of malignant tumors [75]. In addition, it has been reported that Drp1 promotes KRas-driven metabolic changes to drive pancreatic tumor growth and that the expression of dynamin-related protein 1 (Drp1) in epithelial ovarian cancer has a poor prognosis, suggesting that Drp1 may be used as a biomarker for malignant tumors [76,77]. Mitochondrial dysfunction has recently been identified as a pathological factor related to cellular aging [78,79]. Therefore, we hypothesized that the development of Warthin’s tumor may be related to cellular senescence, including Drp1 (Figure 7). IL13 was recently shown to play a critical role in the induction of salivary gland epithelial cell senescence by increasing mitochondrial oxidative stress through a phosphorylated signal transducer and activator of transcription 6 (p-STAT6)-cAMP-response element binding protein (CREB)-binding protein (CBP)-superoxide dismutase 2 (SOD2)-dependent pathway in IgG4-related sialadenitis (IgG4-RS). In addition, a clear increase in SA-β-gal-positive cells in IgG4-RS in both acini tufts and ducts was observed, suggesting the possibility that epithelial cell senescence is present and may be related to salivary gland dysfunction in IgG4-RS [80].

## 8. IgG4 and Warthin’s Tumor

IgG4-RD is a systemic inflammatory disease characterized by severe fibrosis, high serum IgG4 levels (>135 mg/dL), and marked IgG4-positive lymphoplasmacytic infiltrates. It was initially proposed as autoimmune pancreatitis and Mikulicz disease [81,82]. High IgE levels and eosinophilic infiltrates are often observed. However, the etiological mechanisms underlying this IgG4-related disease are largely unknown, and it is unclear whether IgG4-RD is caused by abnormal acquired immunity, such as autoimmune diseases, or whether increased IgG4 production has a direct impact [83]. IgG4 accounts for less than 5% of all IgGs in healthy individuals. IgG4 accounts for a lower percentage in comparison to IgG1 to IgG3, and the Fc region of IgG4 is thought to play a small role in immune activation due to its weak binding to C1q and Fcγ receptors. IgG4 differs from other IgGs after secretion from plasma cells in that the Fab region is exchanged for other Fab regions, allowing a single molecule to recognize different antigens. The resulting antibodies are thought to exhibit anti-inflammatory effects by reducing their ability to form immune complexes [84]. IgG4 production is primarily controlled by Th2 cells [85]. Under antigen stimulation, IgG4 production is induced by IL4 and IL13, which are Th2-type cytokines involved in allergic reactions. In the presence of IL10, IL12, and IL21, IgG4 production becomes predominant over IgE production. In Th2 cytokine-driven immune reactions, IgG4 production is preferentially induced by the activation of IL10 produced by regulatory T (Treg) cells [86]. The overexpression of IL10, TGFβ, and activation-induced cytidine deaminase (AID) has been reported in the labial salivary glands (LSGs) of IgG4-RD patients in comparison to SS patients, suggesting that Treg cytokines (IL10 and TGFβ) combined with AID, an IgG4-unrelated molecule in IgG4-RD (MD) patients, contributes to IgG4-specific class switch recombination and fibrosis [87]. Aga et al. recently suggested an association between Warthin’s tumor and IgG4-RD [48]. In addition, serum IgG4 levels showed an increasing trend in Warthin’s tumors in comparison to pleomorphic adenomas [88].

Our recent findings suggest that Warthin’s tumor tends to have more IgG4-positive cells upon immunohistochemical staining of histological sections, suggesting a certain but not causal relationship between IgG4-RD and Warthin’s tumor (Figure 8 and Figure 9). However, since salivary gland-like cystic carcinomas with IgG4-RD have also been reported recently [89], additional research is necessary to determine whether salivary gland tumors themselves, not just Warthin’s tumors, are prone to producing IgG4 or whether IgG4-positive plasma cell infiltration occurs via tumor-stimulated signals.

## 9. The Role of GAPDH in Warthin’s Tumor Cells

Once considered a simple “housekeeping” protein, the glycolytic enzyme GAPDH has recently been shown to be involved in many cellular functions other than glycolysis [90,91]. In addition, studies pointing to its involvement in apoptosis-promoting functions and tumor progression have suggested that GAPDH depletion is associated with cellular senescence, such as accelerated senescence in tumor cells [92,93,94,95]. However, many aspects of the function of GAPDH remain unclear [96].

Regarding the association between Warthin’s tumor and GAPDH, on anti-GAPDH immunohistochemical staining, Warthin’s tumor cells had a significantly lower percentage of GAPDH-positive cells (*p* < 0.0001) in comparison to normal parotid duct cells [18]. The quantitative analysis of the expression of GAPDH mRNA by quantitative RT-PCR also showed that Warthin’s tumor cells had lower expression levels in comparison to normal parotid duct cells [18]. GAPDH was found to be associated with damaged mitochondria, resulting in direct incorporation of these mitochondria into lysosomes [97]. This suggests that Warthin’s tumor cells gradually lose cytoplasmic GAPDH due to cell-wide GAPDH loss or a nuclear shift [18], as shown in Figure 3.

## 10. The Role of p16 and p53 in Warthin’s Tumor

p16 is a tumor suppressor protein encoded by the cyclin-dependent kinase inhibition 2A (CDKN2A) gene. On immunostaining, p16 positivity is generally a biomarker for HPV infection-related malignancies, such as cervical and pharyngeal cancers. Although this seems contradictory, it is thought to be due to the interference of HPV E6 and E7 viral tumor protein expression with the tumor suppressor p53 and Rb pathways, causing upregulation of p16 expression through the inactivation of Rb by E7 [98,99].

p53 is a tumor suppressor gene that promotes or suppresses the transcription of various proteins, thereby conferring resistance to various cellular stresses. The inhibition of cell cycle progression by p53 is mediated by multiple mechanisms, including the upregulation of p21, which inhibits the cyclin-dependent kinase (CDK) family of kinases, and transcriptional regulation of Gadd45 and 14-3-3σ. p53 maintains genomic stability and is involved in DNA repair, apoptosis, and cellular senescence [100]. The inactivation of p53 tumor suppressor genes occurs frequently during tumorigenesis. In most cases, the p53 gene mutates to produce a stable mutant protein, the accumulation of which is considered a hallmark of cancer cells. Mutant p53 proteins not only lose their tumor suppressor activity, but often acquire additional oncogenic functions that confer growth and survival advantages to the cell [101].

In normal cells, the p53 protein has a very short half-life and is not present in sufficient amounts to be detected by immunostaining. However, abnormal p53 caused by mutations has a long half-life and accumulates in the nucleus. As a result, mutated p53 is easily detectable by IHC. There are interesting studies on the immunohistochemical staining of p16ink and p53 in Warthin’s tumors. Classic cytogenetic studies have identified clonal abnormalities in several Warthin’s tumors [102,103]. Recently, however, several molecular biological studies have begun to address the controversy over whether Warthin’s tumor is neoplastic or nonneoplastic and have shown conflicting results. For example, immunohistochemical staining of p16ink and p53 in 12 cases of Warthin’s tumor was negative in all cases, suggesting that there was no evidence of abnormal staining for tumor suppressor gene protein products (p16ink and p53) and no evidence of consistent clonal allelic deletions, indicating that Warthin’s tumor is non-neoplastic [104]. In another study that examined the clonality of the epithelial component of Warthin’s tumors using a PCR assay based on random inactivation of the gene by trinucleotide repeat polymorphisms and methylation of the X chromosome-related human androgen receptor gene (HUMARA), all cases showed a polyclonal X inactivation pattern, suggesting that Warthin’s tumors are non-neoplastic [105].

In our recent study, immunohistochemical staining of p16 (Figure 10) and p53 (Figure 11) revealed positive nuclei in the columnar cells of Warthin’s tumor, suggesting that Warthin’s tumor is neoplastic, although whether Warthin’s tumor is a true neoplasm that occurs as a clonal growth or a non-neoplastic developmental malformation is still a matter of debate. In addition, our recent unpublished study revealed positive immunohistochemical staining of Ki67/MIB-1 in columnar cells in Warthin’s tumors (Figure 12), indicating that columnar cells possess proliferative activity.

## 11. HPV Infection and Warthin’s Tumor

An interesting study showed that HPV-PCR was performed on 50 of 55 salivary gland tumors that tested positive for p16, and it was found that HPV was not involved in salivary gland pathogenesis [106]. However, in another study, HPV PCR was performed in 25 cases of Warthin’s tumor, and HPV was detected in 19 of 25 cases (76%). The remaining six cases were negative or had no amplified DNA; HPV type 16 was detected in all 19 positive cases that tested positive for HPV by PCR [47]. These results suggest a correlation between the nuclear overexpression of p16 and high-risk HPV infection in Warthin’s tumors. HPV type 16 has also been detected in other salivary gland neoplasms, such as adenoid cystic carcinoma; adenocarcinoma NOS; Warthin’s tumor; and to a lesser extent, acinus cell carcinoma, salivary duct carcinoma, and adenoid basal cell carcinoma. Thus, HPV appears to be involved in a significant proportion of salivary gland tumors, but its exact role remains controversial [47]. It is possible that salivary gland tumors may decrease as HPV vaccines become more widely available. We would propose additional research that can more definitively link HPV to Warthin’s tumor.

## 12. Conclusions

As illustrated in Figure 13, senescent cells and chronic inflammation (SASP) may be associated with the development and progression of Warthin’s tumors. Morphologically, the cytoplasm of Warthin’s tumor reflects a state of mitochondrial overaccumulation called oncocytosis, which is caused by abnormal mitochondrial metabolism and involves damage to the mitochondrial genome. Positive p53 immunohistochemical staining in Warthin’s tumors may reflect mutations in the p53 gene or p53 activity due to cellular senescence. Whether Warthin’s tumor is a true neoplasm that occurs as a clonal growth or nonneoplastic developmental malformation remains controversial. However, given the accumulation of aged mitochondria (senescent cells), HPV positivity, and p53 and p16 positivity in Warthin’s tumor, we believe that Warthin’s tumor is neoplastic.

## Figures and Tables

**Figure 1 cancers-16-00912-f001:**
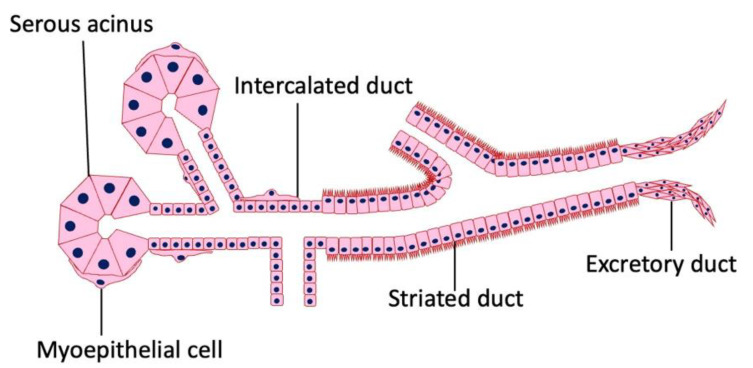
Histology of the salivary gland. Salivary glands are composed of epithelial columnar cells, myoepithelial cells, intercalated duct cells, acinar cells, and connective tissue. There is no lymphatic tissue in the stroma of a normal salivary gland.

**Figure 2 cancers-16-00912-f002:**
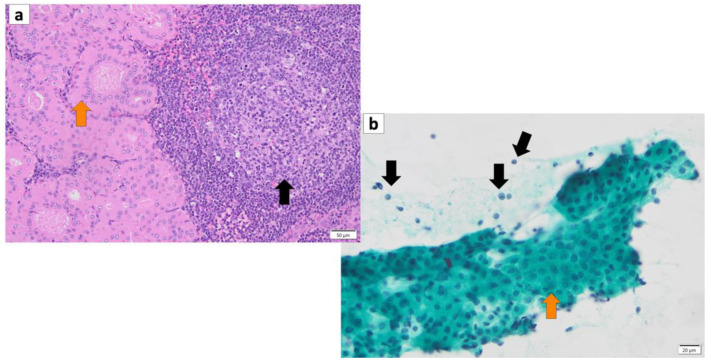
(**a**) Histopathology of Warthin’s tumor. Note: varying proportions of papillary cystic structures are lined with bilayered oncocytic epithelial cells (orange arrow) and surrounded by a lymphoid stroma, including germinal centers (black arrow). (**b**) FNA cytology of Warthin’s tumor shows small cohesive sheets of oncocytes with abundant granular cytoplasm with a central round nucleus/prominent nucleolus (orange arrow). Lymphocytes (black arrows) with granular debris in the background. (**a**) Hematoxylin and eosin staining (Bar represents 50 μm) and (**b**) Papanicolaou staining (bar represents 20 μm).

**Figure 3 cancers-16-00912-f003:**
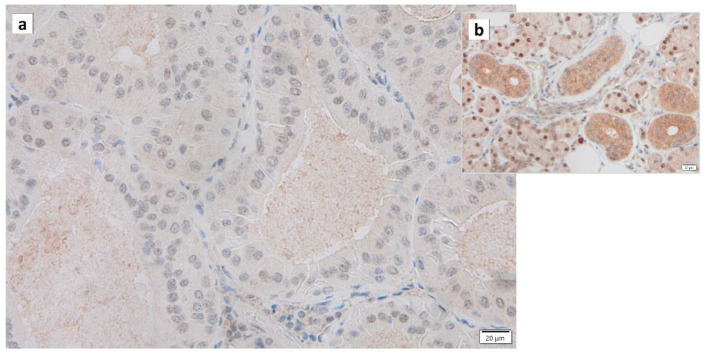
Immunohistochemistry of GAPDH in Warthin’s tumor. Note the (**a**) negative reaction in the columnar epithelial cells and (**b**) positive reaction in the cytoplasm of intercalated ductal cells and some nuclei of acinar cells. Bars represent 20 μm.

**Figure 4 cancers-16-00912-f004:**
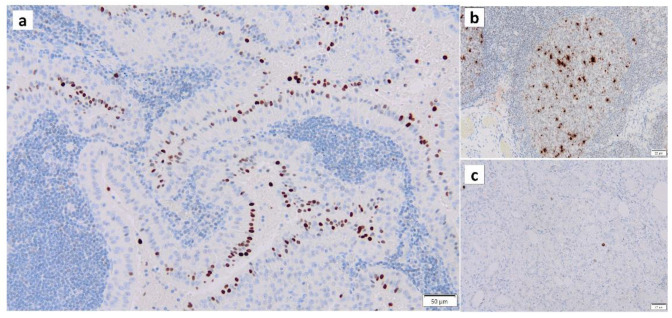
Immunohistochemistry of γ-H2AX in Warthin’s tumor. γ-H2AX positivity is found (**a**) in the nuclei of the columnar epithelial cells and (**b**) the nuclei of lymphocytes in the germinal center. (**c**) The normal salivary gland is negative for γ-H2AX. Bars represent 50 μm.

**Figure 5 cancers-16-00912-f005:**
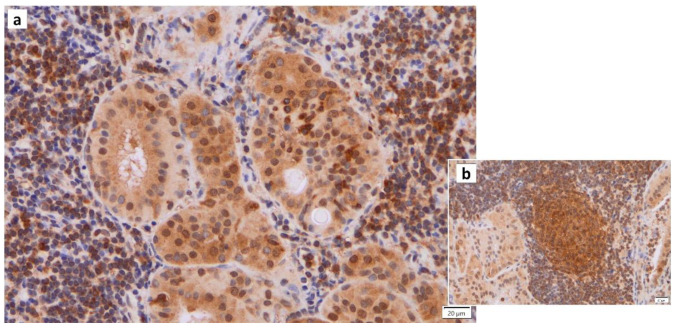
IL1β immunohistochemistry of Warthin’s tumor. (**a**) IL1b positivity is found in some nuclei of the columnar epithelial cells and (**b**) the surrounding lymphocytes in and around the germinal center. Bars represent 20 μm.

**Figure 6 cancers-16-00912-f006:**
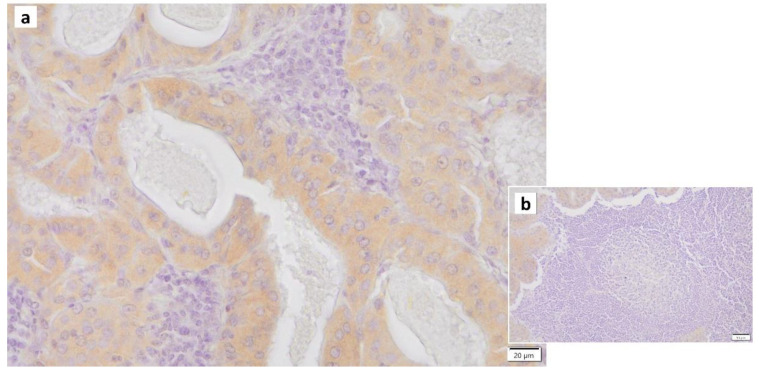
IL6 immunohistochemistry of Warthin’s tumor. (**a**) Weak positivity for IL6 is found in the cytoplasm of the columnar epithelial cells. (**b**) The surrounding lymphocytes in and around the germinal center are negative. Bars represent 20 μm.

**Figure 7 cancers-16-00912-f007:**
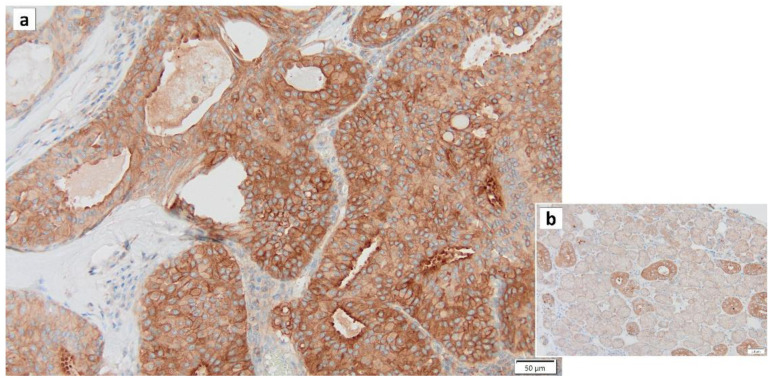
Drp1 immunohistochemistry of Warthin’s tumor. (**a**) Most cytoplasm of the columnar epithelial cells is positive for Drp1. (**b**) In the normal salivary gland, the cytoplasm of intercalated duct cells is also positive for Drp1, while the cytoplasm of acinar cells is negative for Drp1. Bars represent 50 μm.

**Figure 8 cancers-16-00912-f008:**
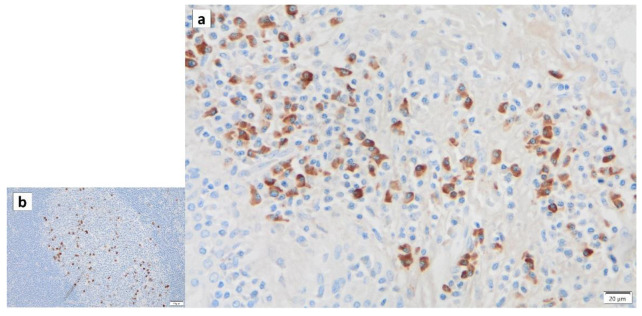
IgG4 immunohistochemistry of Warthin’s tumor. IgG4 positivity is detected in the lymphocytes of (**a**) the surrounding tissue and (**b**) the germinal center. Bars represent 20 μm.

**Figure 9 cancers-16-00912-f009:**
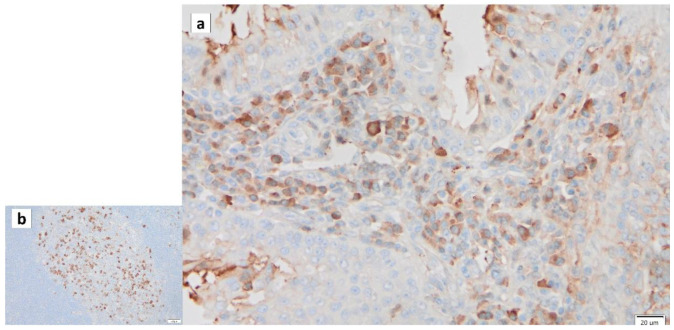
IgG immunohistochemistry of Warthin’s tumor. IgG is detected in (**a**) some of lymphocytes in the surrounding stroma and (**b**) germinal center. Bars represent 20 μm.

**Figure 10 cancers-16-00912-f010:**
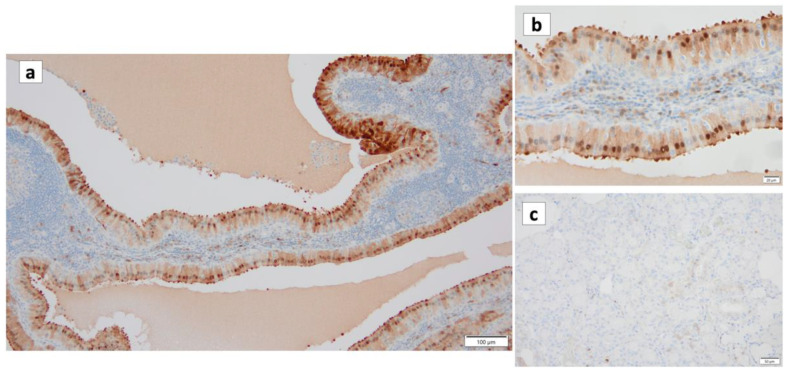
p16 immunohistochemistry of Warthin’s tumor. (**a**,**b**) p16 immunoreactivity is noted in the nuclei of the columnar cells and some surrounding lymphocytes. (**c**) On the other hand, p16 is not detected in the normal salivary gland. (**a**) Bar represents 100 μm. (**b**) Bar represents 20 μm. (**c**) Bar represents 50 μm.

**Figure 11 cancers-16-00912-f011:**
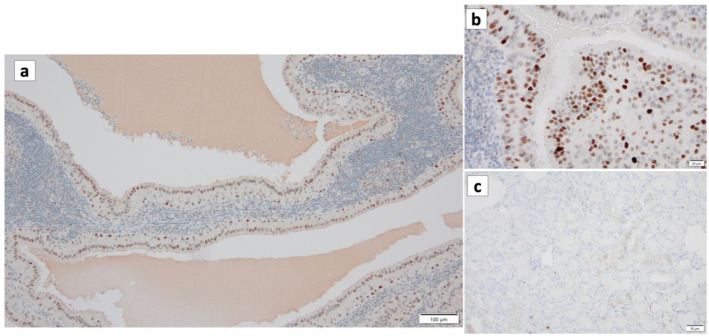
p53 immunohistochemistry of Warthin’s tumor. (**a**,**b**) p53 immunoreactivity is noted in the nuclei of the columnar cells and some surrounding lymphocytes. (**c**) However, p53 is not detected in the normal salivary gland. (**a**) Bar represents 100 μm. (**b**) Bar represents 20 μm. (**c**) Bar represents 50 μm.

**Figure 12 cancers-16-00912-f012:**
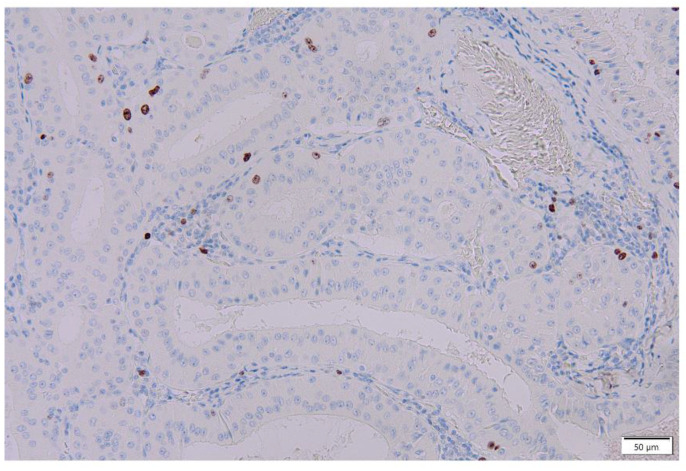
Ki67/MIB-1 immunohistochemistry of columnar cells in Warthin’s tumors. Several nuclei of the columnar cells in the Warthin’s tumor are positive for Ki67/MIB-1. Bar represents 50 μm.

**Figure 13 cancers-16-00912-f013:**
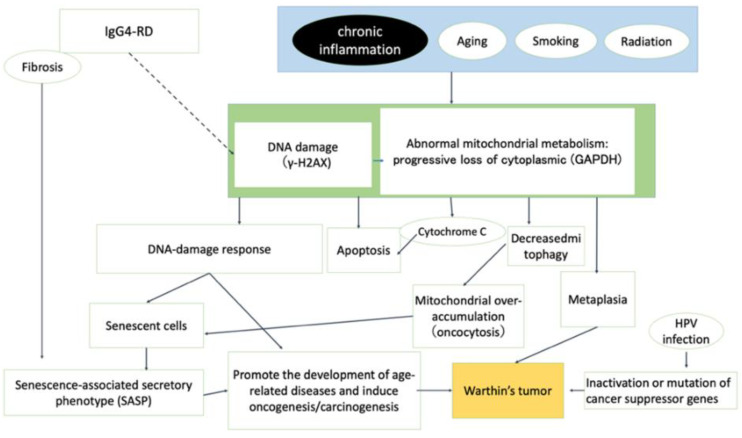
Graphical summary of the different factors that contribute to the development of Warthin’s tumor.

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
