# Peer review of "Pathogenesis of Warthin’s Tumor: Neoplastic or Non-Neoplastic?"

_cancers, 2024, doi:10.3390/cancers16050912_

Round 1
Reviewer 1 Report
Comments and Suggestions for Authors
Thank you for the submission.
The authors failed to give the non-eponymous name for Warthin's tumors: papillary cystadenoma lymphomatosum. It is worth mentioning in the introduction.
Regarding the following sentence: "Our unpublished data also showed that Warthin’s tumor tended to have more IgG4- positive cells on immunohistochemical staining of histological sections, suggesting a causal relationship between IgG4-RD and Warthin’s tumor (Figures 8 and 9)." I don't believe you can assume a CAUSAL relationship with the data you report. Please modify the language of this sentence to not overstate your findings/conclusions about IgG4.
I would like the authors to comment on the negative HPV p16 studies. If they believe HPV to be a risk factor for salivary gland tumors and Warthin's tumors, the authors should propose an explanation for the negative studies or propose additional research that can more definitively link HPV to Warthin's tumors.
Overall, there is a wealth of scientific information in this submission that is valuable and well presented. The figures are particularly helpful.
Comments on the Quality of English Language
some minor English corrections are required.
Author Response
Reviewer 1
Thank you very much for carefully reviewing our manuscript and for the valuable and positive comments for us to revise the manuscript.
1. The authors failed to give the non-eponymous name for Warthin's tumors: papillary cystadenoma lymphomatosum. It is worth mentioning in the introduction.
Reply: Thank you for your suggestion. The term "papillary cystadenoma lymphomatosum" has been added
2. Regarding the following sentence: "Our unpublished data also showed that Warthin’s tumor tended to have more IgG4- positive cells on immunohistochemical staining of histological sections, suggesting a causal relationship between IgG4-RD and Warthin’s tumor (Figures 8 and 9)." I don't believe you can assume a CAUSAL relationship with the data you report. Please modify the language of this sentence to not overstate your findings/conclusions about IgG4.
Reply: We agree with you. We have modified the sentence as you indicated.
3. I would like the authors to comment on the negative HPV p16 studies. If they believe HPV to be a risk factor for salivary gland tumors and Warthin's tumors, the authors should propose an explanation for the negative studies or propose additional research that can more definitively link HPV to Warthin's tumors.
Reply: Many thanks for your suggestion. We have added a new sentence.
Reviewer 2 Report
Comments and Suggestions for Authors
The authors concentrated on a popular concer in salivary gland tumors, that is Worthin's tumor is a neoplasm or non-neoplasm. By summarizing the previous studies and their own findings, the authors concluded the neoplastic identity of Worthin's tumor. This review would shed light on the study on the genesis of Worthin's tumor.
The most concern on this manuscript is confusing organization and format of the mansucript. As a review, the present in a format of research article. As a resarch article, it failed to give hypothesis and test it through results.
I suggest the authors re-organized the masnucript with explicating sub-topics, and insert their own findings into the corresponding location as the cited references to construc an integrated network.
Comments on the Quality of English LanguageThe english language still requires progress, because there are a lot of "the" in the inapproprated position. Additionally, some sentences are confusing, such as "Previously, tumors were considered more common in men in their 60s, and women have been increasingly affected inrecent years, indicating that Warthin’s tumor may surpass pleomorphic adenoma, the most frequently occurring salivary gland tumor".
Author Response
Reviewer 2
Thank you very much for carefully reviewing our manuscript and for the valuable and positive comments for us to revise the manuscript.
1. The most concern on this manuscript is confusing organization and format of the manuscript. As a review, the present in a format of research article. As a research article, it failed to give hypothesis and test it through results.
Reply: Subheading “Materials and Methods” has been deleted, as you recommended.
2. I suggest the authors re-organized the manuscript with explicating sub-topics, and insert their own findings into the corresponding location as the cited references to construct an integrated network.
Reply: Thank you for your kind advice. We have slightly modified the text.
3. The English language still requires progress, because there are a lot of "the" in the inapproprated position. Additionally, some sentences are confusing, such as "Previously, tumors were considered more common in men in their 60s, and women have been increasingly affected inrecent years, indicating that Warthin’s tumor may surpass pleomorphic adenoma, the most frequently occurring salivary gland tumor".
Reply: Thank you for your kind advice. English of this manuscript was edited by Professor who is a native speaker and familiar with Medicine. The sentences you indicated have been modified.
Reviewer 3 Report
Comments and Suggestions for Authors
Dear Authors,
I found your review discussing various factors, which participate in Warthin's tumor development and progression, very interesting (especially concerning the original data presented), well structured and written in good English. My comments concern only several typos I noticed:
1. Figure 1 a - bar is 100 mkm while in caption stated 20 mkm.
2. Page 9, line 2 - 135 mg/day (Is it correct?)
3. Page 10, section 2.8, line 3 - dot before references.
Author Response
Reviewer 3
Thank you very much for carefully reviewing our manuscript and for the valuable and positive comments for us to revise the manuscript.
1. Figure 1 a - bar is 100 mkm while in caption stated 20 mkm.
Reply: As you suggested, we have corrected Bars.
2. Page 9, line 2 - 135 mg/day (Is it correct?)
Reply: Page 9, line 2 -We have corrected our mistake “135 mg/day” to “135 mg/dL”.
3. Page 10, section 2.8, line 3 - dot before references.
Reply: We have corrected our mistake.
Reviewer 4 Report
Comments and Suggestions for Authors
This article establishes a comprehensive analysis of different IHC markers expression in Warthin's tumor. Although the main idea of this review was to answer the question about neoplastic nature of this lesion. I suppose that IHC data is not sufficient to give the answer. The authors should analyse experimental and in vitro data or reoriented this review for pathologists provided more clinical data and insert a piece of information about differential diagnoses. In addition the final figure looks careless so I recommend the authors to make it more accurately.
Author Response
Reviewer 4
Thank you very much for carefully reviewing our manuscript and for the valuable and positive comments for us to revise the manuscript.
This article establishes a comprehensive analysis of different IHC markers expression in Warthin's tumor. Although the main idea of this review was to answer the question about neoplastic nature of this lesion. I suppose that IHC data is not sufficient to give the answer. The authors should analyze experimental and in vitro data or reoriented this review for pathologists provided more clinical data and insert a piece of information about differential diagnoses. In addition, the final figure looks careless so I recommend the authors to make it more accurately.
Reply: We agree with you, However, we have no equipment for molecular analysis. At present, our weapon is IHC. Firstly, we have tried IHC analysis, and then we will perform molecular analysis in the near future after getting equipment and technique of molecular analysis. We believe that experimental data using genomes in addition to IHC will be necessary in the future. Final figure has been modified.
Reviewer 5 Report
Comments and Suggestions for Authors
This is an interesting review of histopathology with clear explanations of possible mechanisms of Warthin’s tumors development and characteristics. The authors at the end propose that contrary to the current view, that Warthin’s tumors are neoplastic. This reviewer’s major points are that histology figures need to be presented more clearly with arrows and legends that identify features of interest. Second, some primary data are presented and could be improved by demonstrating co-staining of the same cells of p53, p16, and Ki67. In addition, a few clarifications in the writing are needed. All of this is indicated in specific comments below:
1. Figure 2a. Arrows and possibly higher magnification are needed to help to orient the reader to the histopathology of Warthin’s tumors.
2. Figure 2b. Higher magnification with arrows pointing to specific features of interest are needed.
3. Figures 3 - 7. Here again arrows identifying specific cell types and features are required for readers to fully appreciate the histopathology.
4. Are any genes or regulatory sequences encoded in the commonly deleted 4877 bp deleted region of mitochondrial DNA?
5. The following quoted sentence strongly suggests a causal relationship, while data presented are correlative only; without further data the sentence should be revised. “Our unpublished data also showed that Warthin’s tumor tended to have more IgG4- positive cells on immunohistochemical staining of histological sections, suggesting a causal relationship between IgG4-RD and Warthin’s tumor (Figures 8 and 9).”
6. What is the difference between Figure 8 and Figure 9, and what point is being made by presenting both of these figures?
7. The statement that GAPDH has no catalytic activity is incorrect. Is this what the authors mean to convey? Please explain.
8. In the authors’ primary data Figures 10 and 11, did the authors investigate if p53 and p16 were expressed in the same cells? Figure 12 would benefit from showing the degree of staining for Ki67 in normal salivary glands. Co-staining in the same cells would provide stronger support for the hypothesis that Warthin’s tumors can be neoplastic. The authors are free to present their hypothesis, but may need to explain what may be missing in the data that they present.
Author Response
Reviewer 5
Thank you very much for carefully reviewing our manuscript and for the valuable and positive comments for us to revise the manuscript.
1) Figure 2a. Arrows and possibly higher magnification are needed to help to orient the reader to the histopathology of Warthin’s tumors.
Reply: Figure 2a has been modified with a higher magnification image and arrows.
2) Figure 2b. Higher magnification with arrows pointing to specific features of interest are needed.
Reply: We have added arrows to indicate lymphocytes and epithelial cells.
3) Figures 3 - 7. Here again arrows identifying specific cell types and features are required for readers to fully appreciate the histopathology.
Reply: We have not added arrows in Figures 3~7, because the images are easy to understand.
4) Are any genes or regulatory sequences encoded in the commonly deleted 4877 bp deleted region of mitochondrial DNA?
Reply: "mtDNA4977 common deletion" eliminates between nucleotides 8470 and 13447 of the human mitochondrial genome. mtDNA4977 removes all 5 tRNA genes (tRNAGly, tRNAArg, tRNAHis, tRNASer and tRNALeu) and 7 genes encoding 4 Complex I subunits (ND3, ND4, ND4L, partial ND5), 1 Complex IV subunit (COX III), 2 Complex V subunits (ATP6 and partial ATP8), that are crucial for supporting normal mitochondrial OXPHOS function. Please read a paper (PMID:31044027).
5) The following quoted sentence strongly suggests a causal relationship, while data presented are correlative only; without further data the sentence should be revised. “Our unpublished data also showed that Warthin’s tumor tended to have more IgG4- positive cells on immunohistochemical staining of histological sections, suggesting a causal relationship between IgG4-RD and Warthin’s tumor (Figures 8 and 9).”
Reply: We have modified the sentence, as you suggested.
6) What is the difference between Figure 8 and Figure 9, and what point is being made by presenting both of these figures?
Reply: Figure 8 is an IgG4 and Figure 9 is an IgG IHC image, because when diagnose IgG4-RD, the IgG4/IgG positive cell ratio is quite important and should be included as a part of the criteria.
7) The statement that GAPDH has no catalytic activity is incorrect. Is this what the authors mean to convey? Please explain.
Reply: We have corrected our mistake you suggested.
8) In the authors’ primary data Figures 10 and 11, did the authors investigate if p53 and p16 were expressed in the same cells? Figure 12 would benefit from showing the degree of staining for Ki67 in normal salivary glands. Co-staining in the same cells would provide stronger support for the hypothesis that Warthin’s tumors can be neoplastic. The authors are free to present their hypothesis, but may need to explain what may be missing in the data that they present.
Reply: In the IHC images in Figures 10 and 11, p53 and p16 were re-photographed by low and high magnifications, as recommended.
Reviewer 6 Report
Comments and Suggestions for Authors
The article “Pathogenesis of Warthin’s tumor: Neoplastic or non-neoplastic?” is very interesting, but I will make some comments with the intention of improving it:
- The material and methods section should not be included in the review such as subheading.
- Anatomy: rectify the phrase “Three types of ducts are present in all of the salivary glands”. There are salivary glands that do not have any type of duct.
- Indicate the sensitivity and specificity of fine-needle aspiration cytology in the diagnosis of WT.
- Omit unpublished personal opinions, such as “However, based on our unpublished data and previous studies, we believe that mitochondrial metabolic abnormalities, senescent cell accumulation, chronic inflammation, senescence-associated secretory phenotype (SASP), and HPV infection may be involved in the pathogenesis of Warthin's tumor” or “Our unpublished data also showed that Warthin's tumor tended to have more IgG4-positive cells on immunohistochemical staining of histological sections, suggesting a causal relationship between IgG4-RD and Warthin's tumor (Figures 8 and 9).
- The authors should provide references related to chronic inflammation and tumorigenesis of salivary glands or head and neck cancer rather than colon cancer.
- Comment on the results of the investigation by Kuzenko et al.
Thank you
Author Response
Reviewer 6
Thank you very much for carefully reviewing our manuscript and for the valuable and positive comments for us to revise the manuscript.
1. The material and methods section should not be included in the review such as subheading.
Reply: As you indicated we have deleted the subheading “Materials and methods”.
2. Anatomy: rectify the phrase “Three types of ducts are present in all of the salivary glands”. There are salivary glands that do not have any type of duct.
Reply: As you indicated we have corrected our mistake.
3. Indicate the sensitivity and specificity of fine-needle aspiration cytology in the diagnosis of WT.
Reply: As you recommended, we have added the sensitivity and specificity of FNA cytology in the diagnosis of WT.
4. Omit unpublished personal opinions, such as “However, based on our unpublished data and previous studies, we believe that mitochondrial metabolic abnormalities, senescent cell accumulation, chronic inflammation, senescence-associated secretory phenotype (SASP), and HPV infection may be involved in the pathogenesis of Warthin's tumor” or “Our unpublished data also showed that Warthin's tumor tended to have more IgG4-positive cells on immunohistochemical staining of histological sections, suggesting a causal relationship between IgG4-RD and Warthin's tumor (Figures 8 and 9).
Reply: As you suggested, we have modified the sentences including “our unpublished”.
5. The authors should provide references related to chronic inflammation and tumorigenesis of salivary glands or head and neck cancer rather than colon cancer.
Reply: We have replaced Ref. no. 65 by new one (PMID: 31473936).
6. Comment on the results of the investigation by Kuzenko et al.
Reply: We have added the comment on an interesting study by Prof. Kuzenko.
Round 2
Reviewer 2 Report
Comments and Suggestions for Authors
All concerns have been addressed. No more question.
Reviewer 4 Report
Comments and Suggestions for Authors
The authors modified the manuscript as far as they could do it with their poor equipment. So I suppose that the article could not be improved without addition investigations such as molecular and experimental which the authors are planning to perform in near future.